# Absolute Environmental Sustainability of Materials Dissipation: Application for Construction Sector

Wafaa Baabou [1,2,*], Anders Bjørn [3,4] and Cécile Bulle [1,2]

1   CIRAIG, Department of Strategy and Corporate Social Responsibility, ESG UQAM, Montréal, QC H2X 3X2, Canada
2   Environmental Sciences Institute, UQAM, Montréal, QC H2L 2C4, Canada
3   Department of Management, John Molson School of Business, Concordia University, 1450 Guy St, Montréal, QC H3H 0A1, Canada
4   Department of Geography, Planning and Environment, Concordia University, 1455 de Maisonneuve Blvd. W, Montréal, QC H3G 1MB, Canada
*   Correspondence: baabou.wafaa@courrier.uqam.ca

**Abstract:** The materials used globally in the construction sector are projected to more than double in 2060, causing some to deplete. We argue that access to the services that the resources provide must be protected, thus implying that a carrying capacity (CC) for resource dissipation must be set. Dissipation accrues when the resource becomes inaccessible to users. The CC allows defining a maximum dissipation rate that allows to maintain those resources' availability in the future. The CC of the dissipation of the resource may be operationalized to characterize the resource use impact, using absolute environmental sustainability assessments principles. The study makes it possible to determine a dissipation CC as the world dissipation rate that would enable all users to adapt to using an alternative resource before the material's reserve is entirely dissipated. The allocation of a fraction of this CC to the building sector was performed using equal per capita and grandfathering sharing principles. Finally, we applied the method to the case of steel in a school life cycle. The results show that the actual dissipation rates of iron, copper and manganese in the building sector exceed the dissipation CC by 70%, 56% and 68%, respectively. However, aluminum dissipation is 90% less than the assigned CC. The allocation to schools shows that the results are influenced by the choice of allocation principle. The application in the case of steel use of the school life cycle shows an exceedance of the CC that decreases when increasing the building life span.

**Keywords:** construction materials; schools; dissipation; user adaptation; carrying capacity; allocation approaches

## 1. Introduction

Global primary materials use and, by extension, global primary materials extraction is projected to double in the coming decades from 79 gigatons (Gt) in 2011 to 167 Gt in 2060 [1]. The construction sector is projected to more than double between 2017 and 2060 leading to almost 84 Gt of construction materials used per year in 2060. In the Canadian context, the construction sector consumes 50% of natural resources and contributes to 11% of national $CO_2$ emissions (Environment Canada, 2014). The construction sector is therefore among the most impactful in terms of greenhouse gases and natural resource depletion.

Life cycle assessment (LCA) methods are commonly used in environmental assessments of building sector sustainability to support decision making and evaluate and optimize construction processes and building operation. Life cycle impact assessment (LCIA) methods make it possible to quantify potential environmental impacts, including abiotic resource use. However, while there is a general consensus on how to characterize impact categories, such as climate change [2] or toxic impacts [3], there is no global consensus on the assessment of abiotic resource depletion [4]. One of the main issues

with the impact assessment of resource depletion is the clear definition of the area of protection (AoP): what do we want to protect by avoiding abiotic resource depletion? While the human health and ecosystem quality AoPs protect life, which has an intrinsic value, resources are considered to have an instrumental value [5,6]. The LCA community has been developing new but divergent methods that all focus on different issues related to resource use [4,7]. A first category of methods to assess resource use focuses exclusively on extraction from the environment and compares it to the available geological stock. For example, the abiotic depletion potential (ADP) [8] assesses the extraction of a resource, considering its geological reserve compared to a reference resource (antimony). Another category of methods also considers secondary resources by taking into account anthropogenic stock in addition to geological stock. For example, the extended abiotic depletion potential [AADP] [9] considers that the abiotic resources may remain available for further uses in the anthropogenic system. From this perspective, the dissipation, rather than the extraction, may be considered the cause of resource depletion [10]. A third category of method quantifies the reduction in exergy, known as the cumulative exergy extraction from the natural environment (CEENE) [11]. A final category of LCIA methods is based on a function-based approach model [12] and has been operationalized in the MACSI model (material competition scarcity index) [13] that characterizes the fraction of users who are unable to adapt before the easily accessible reserves are projected to be fully depleted. This competition evaluation model is based on two parameters: the time before depletion (which is a ratio of accessible reserves over the annual dissipation rate of resources) and the potential adaptation time of users facing resource depletion.

Access to the services and functions that a resource provides may be ensured by keeping the dissipation rate within a certain limit. This limit is conceptually similar to the carrying capacity (CC) used in LCA-based absolute environmental sustainability assessment (AESA), which evaluates whether an anthropogenic system may be considered environmentally sustainable in an absolute sense [14]. According to [14], the overall framework of the LCA-based AESA is constituted of four steps: estimate the environmental impact; quantify the CC (or emission budget, impact budget or safe operating space); assign CC to an anthropogenic system using one or more allocation principles and assess the absolute sustainability of the anthropogenic system by comparing the environmental impact to the assigned CC. Considering the instrumental value of resources as what has to be protected, a CC may be defined as the rate of dissipation that enables all users to adapt before all the easily available stock (geological and anthropic) is depleted. In this context, [15] defined the CC simply as the total quantity of existing accessible geological reserves divided by a specified time horizon, which is considered the time needed for all users of all resources to adapt to other alternatives (arbitrarily set to 200 years). Ref. [16] used the adaptation time to calculate an optimal extraction rate and a reduction factor (RF) for the extraction of each resource. The study then used those RF as weighting factors for characterization factors from CML. Unlike the method by [15] that uses a common, arbitrary time horizon, the method by [16] uses an adaptation time that differs from one resource to another based on substitutability and end-use sharing data. Ref. [17] defined a global CC for material use as the quantity of new mineral resources made available by technological improvement on an annual basis. She therefore considered that the carrying capacity depends only on technological development.

Although the literature progresses in defining the CC of abiotic resources, there are some gaps, such as considering the extraction rather than dissipation, and the lack of considering futures changes in demand. In fact, both of those factors influence the extraction of the resource and thus the CC.

The allocation of CC to specific anthropogenic systems (nations, sectors, companies and products) has been applied by different authors using different sharing principles, which strongly influences the conclusions on whether the activity is considered sustainable or not [18].

At the scale of the built environment, AESAs have been carried out by a number of publications. A detailed summary of studies using allocation principles to build the sector is presented in the Appendix D. For example, ref. [19] assigned the global carbon budget to New Zealand using the cumulative population. The grandfathering sharing principle was used to assign a share of New Zealand's carbon budget to the New Zealand detached housing sector. LCA was then used to separate house building per life cycle stage. Ref. [20] assessed the absolute sustainability of Danish dwellings through a case study, using equal per capita allocation. The study allocated a budget to buildings according to an economic allocation and grandfathering. For non-residential buildings, ref. [21] developed a simplified tool that allocates climate change budgets for colleges and universities and calculates a context-based carbon score that represents the ratio of the university's actual emissions to its fair share of the carbon budget based on the time student and employees spend in the college or university and considering an equal per capita allocation principle. Ref. [18] applied multiple allocation approaches to study the absolute sustainability of diverse dwelling scenario life cycle. Nevertheless, the allocation of carrying capacity to the building sector was not applied in the literature for the case of the resource use impact category.

Although there is a rising literature that addresses the AESA of building sector for different LCA impact categories, the impact of resource use has not been covered yet. In this context, critical questions remain: 1—How can we define a global carrying capacity based on the resource dissipation and functionality? 2—How do we carry out the absolute environmental sustainability assessment (AESA) of buildings life cycle based on the estimated CC?

In this study, we answer those questions through a methodology that allows estimating a carrying capacity, and then applying the concept of AESA to building for this impact category. We consider a Canadian elementary school as a building type. The remainder of the article is structured into four sections. Section 2 presents the method used to calculate the carrying capacity (CC) of abiotic resource dissipation and sharing principles applied to allocate a dissipation for Canadian schools. Section 3 reports the results, including national CC and allocated budgets, as well as a sensitivity analysis for different allocation approaches, and discusses the results and limitations of the analysis. Finally, Section 4 draws the overarching conclusions.

## 2. Materials and Methods

Figure 1 indicates the method built on the framework for LCA-based AESA developed by [14]. The first step of the framework is estimating the potential impact of the use of materials, which we consider here as the dissipation per year (the amount that is not recoverable). The second step is to determine the CC for the dissipation of a material, and the third is to then assign the CC to an anthropogenic system. The method builds on the MACSI model [13]. In this paper, the carrying capacity is defined as the maximum annual dissipation rate required to maintain the functions of a material until the point in time when the fraction of the user not adapted is zero or, in other words, maintaining the reserve in a such way that the time of adaptation ($t_{adapt}$) is equal to the time of depletion (Ddi). The study covers abiotic resources for which reserve data are available [22]: iron, aluminum, manganese, copper, diatomite, perlite and molybdenum. The annual carrying capacity is then allocated to the building sector and more specifically to schools as a type of building by means of a combination of two allocation approaches: equal per capita and grandfathering (detailed below). A sensitivity analysis is undertaken, involving two additional allocation approaches: final consumption expenditures (FCE) and economic value added (EVA). We focus on the use of steel during the school life cycle to conduct an assessment of absolute sustainability for the metal.

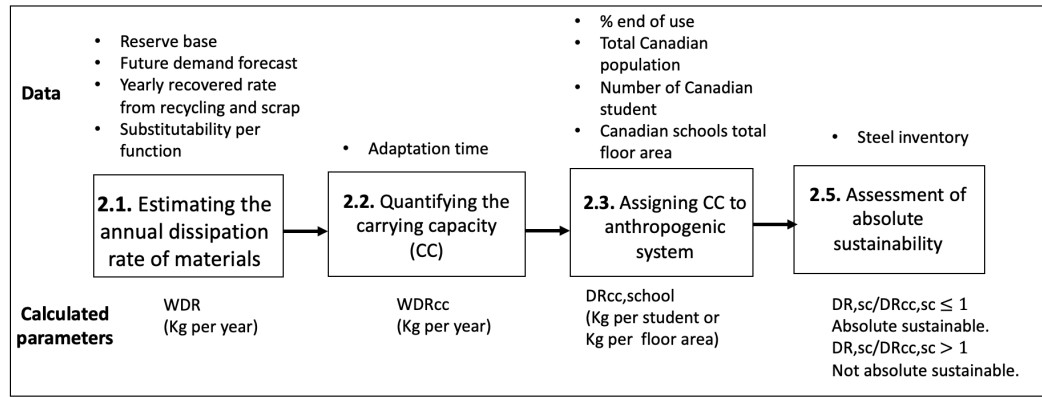

**Figure 1.** Overview of method steps. The numbers refer to sections in this study.

### 2.1. Estimating the Annual Dissipation Rate of Materials

To apply the MACSI method, we must consider the data of the reserve base as the easily available geological stock, world production rate, and world recovery rate (recycling + reuse). The reserve base is defined as part of an identified resource that has a reasonable potential for becoming economically available with planning horizons [22]. It is a reasonable compromise between considering 100% of the Earth crust content as being available for extraction and only the currently available economic stock as being available for extraction. However, the estimation of the reserve base by the US Geological Survey (USGS) was discontinued in 2010, and we therefore had to generate our own estimations for the subsequent years. The reserve base for the year t $WRB_t$ was estimated based on the reserve base of the previous year $WRB_{t-1}$ and the world resource recovery rate of the previous year $(t-1)$ $WDR_{t-1}$ (Equation (1)).

$$WRB_t = WRB_{t-1} - WDR_{t-1} \tag{1}$$

We replaced *WDR* with the world recovery rate of the material ($WRR_t$) plus the annual world production rate $WPR_t$ (Equation (2)). The world recovery rate encompasses the world recycling rate and quantifies all the other materials that enter the economy from secondary sources (i.e., including recycling and reuse). *WDR* is estimated for all years until the dissipation of the reserve base ($WRB_t = 0$).

$$WRB_t = WRB_{t-1} - WPR_{t-1} + WRR_{t-1} \tag{2}$$

#### 2.1.1. Estimating the Future Production of Construction Materials

The economic growth implies higher materials use, especially for construction materials and metals. Similarly, population growth boosts materials consumption. Beyond population and economic growth, the production of materials may also be affected by changes in prices and technology developments. However, those last factors are not easily quantifiable. In this context, multiple studies forecast the demand for materials based on the shared socioeconomic pathways (SSPs) of the UN [23–25]. The SSPs may be considered in a top-down approach to estimate the global demand for materials without studying sector development. Building on [26], we used the IPAT equation (see Appendix A) that expresses I, the environmental impact (production) as a function of P, population, A, affluence (GDP per capita) and T, technology (impact per GDP). A regression analysis based on historical production, population and GDP (a sample of past 20 years) data makes it possible to estimate C (constant) and coefficients b and c. We chose to use the prediction of GDP and population growth based on the medium socio-economic pathway (SSP) [27]. In this case, the global population is expected to reach 8.5 billion in 2030, 9.7 billion in 2050 and 10 billion in 2100 [25,28]. We then calculated the $WPR_{(projected)}$ based on the other SSPs to assess the sensitivity of the estimated demand to the SSP.

### 2.1.2. Estimation of User Adaptation

The fraction of non-adapted users $\eta_i$(t) (Equation (3)) is the fraction of users that still need to use the resource function when the reserve base is entirely depleted and requires a back-up technology to extract the resource at a higher price [13].

$$\eta_i(t) = 1 - t/t_{adapt,i} \tag{3}$$

The adaptation time ($t_{adapt}$) is the time it takes users of a resource's function to substitute this function by another alternative. It depends on the availability of alternatives, their cost and their performance. Ref. [29] identified four cases with different substitutability scores $\sigma$ for functionality i: the resource is non-substitutable ($\sigma_i = 1$, $t_{adapt}$ 1000 years), the resource is substitutable at high cost and with performance loss ($\sigma_i = 0.7$, $t_{adapt} = 240$ years), the resource is substitutable at a low cost ($\sigma_i = 0.3$, $t_{adapt} = 35$ years) and the resource is easily substitutable and other alternatives are already available to fulfill the same function ($\sigma_i = 0$, 5 years). The different adaptation times are estimated based on $\sigma_i$. De Bruille (2014) assumed that $t_{adapt}$ is infinite for non-substitutable functionality, and $t_{adapt}$ is five years for fully substituted functionality. We assume that all users start to adapt at a time of 50 years before the entire dissipation of the resource. This value means that users will notice the need to shift to other substitutes as available reserves decrease. The fraction of users adapted ($1-\sigma_i$(t)) each year is then translated to an approximate quantity ($Q_t$) (Equation (8)) of materials that are substituted and subtracted from the world production rate ($WPR_t$).

$$Q_t = 1 - \eta_i(t) \times WPR_t \tag{4}$$

$$WPR_t = WPR_{projected} - Q_t \tag{5}$$

### 2.2. Quantifying Carrying Capacity

$WDR_{CC,t}$, the carrying capacity in this study, represents the maximum dissipation rate that a material should not exceed annually to maintain the reserve until the fraction of the non-adapted user is equal to 0 (Equation (6)). To calculate this annual maximum dissipation, we explore the different adaptation times (Appendix B) for the multiple functions fulfilled by the material. The higher adaptation time is then used to calculate the annual production rate that must not be exceeded in order to maintain reserves until the entire dissipation of the $WRB$. $WPR_t$ is obtained in Section 2.1.1.

$$WDR_{CC,t} = Dt \times WRB \quad \text{Where} \quad Dt = WPR_t / \sum_{tadapt}^{t0} WPR_t \times 100 \tag{6}$$

### 2.3. Assigning Carrying Capacity to an Anthropogenic System

Figure 2 represents the general methodology used to assign dissipation to the building sector and to Canadian schools by exploring two different allocation approaches.

To assign the global dissipation rate carrying capacity at an individual level, the sharing principle equal per capita (EPC) is applied (Equation (7)). We then use the Canadian population $POP_t$ to estimate a $WDR_{CC}$, per capita at the Canadian level.

$$WDR_{CC,percap} = WDR_{CC,t} / POP_t \tag{7}$$

As each material fulfills multiple functions, including those in the building construction sector, we share the assigned carrying capacity according to the end of use percentage $\delta_i$ (or the percentage of the use of resource x in the construction sector) (Equation (8)). This assignment is known as grandfathering (GF), as the end of use $\delta_i$ reflects the inherited share of building construction to the overall historical dissipation of a material.

$$WDR_{CC,per.cap,B} = WDR_{CC,per.cap} * \delta_i \tag{8}$$

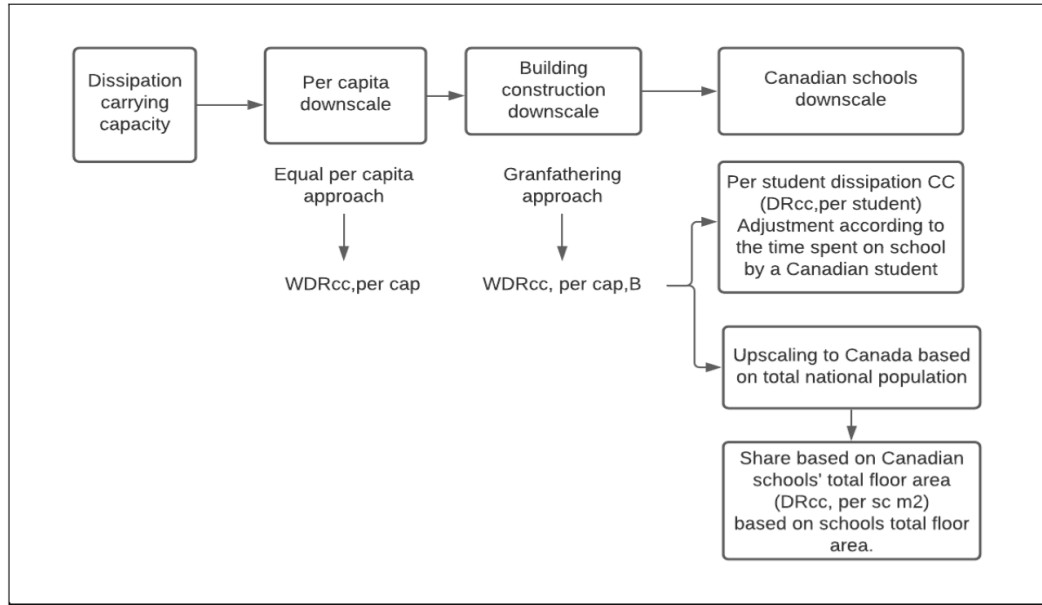

**Figure 2.** Overview of the applied allocation approaches to schools.

The building sector contains all types of building (residential, hospitals, hotels, universities, schools, etc.). To share the *CC* between building types, we scale the $WDR_{CC percap,B}$ per student (kg/student) or per square meter of floor area (kg/m$^2$). For the assignment per student, we used an equal per capita allocation of all the CC of each resource, then applied a grandfathering principle to calculate a per capita CC for the building sector. We then considered the time that an average student spends at school annually to allocate the share to the school building. For the assignment per square meter of floor area, we used the contribution of Canadian schools to the total floor space of all Canadian buildings. Using those methods, the total CC for a school was obtained based on its student headcount or total floor area.

### 2.3.1. Estimation of the Amount of Time Students Spend on School

Equation (9) assigns a fraction of CC to schools. Students do not spend 100% of their time in school, and that should be considered to avoid generating an inflated allocation. In Canada, the time students spend in school annually varies between provinces, with an average of 6 hours per day multiplied by the number of instruction days (187) in a year [30]. The approximate fraction of time that a student spends on school in a year is therefore 0.13.

$$WDR_{CC,per.st} = WDR_{CC,per.cap,B} * 0.13 \tag{9}$$

where $WDR_{CC,perst}$ is the carrying capacity per student and $WDR_{CC,per.cap,B}$ is the carrying capacity per capita for the building sector.

### 2.3.2. Estimation of a School's Total Floor Area

We explored another alternative to downscale CC to a school, which is the allocation per m$^2$ of the school. Equation (10) is used to calculate the dissipation rate per square meter of a school (DR$_{CC,perm^2}$). According to [31], the share of the school's floor space to the Canadian building floor space is 10.9%, which represents 83.6 million square meters of floor area. The number and therefore the total floor area may change but we assume that the share remains the same and proportional to the school's area.

$$DR_{CC,perm^2} = DR_{CC,Canada} * (10.9\%/83.6 \times 10^{+6}) \tag{10}$$

where $DR_{CC,Canada}$ represents the total carrying capacity at the national level.

*2.4. Sensitivity Analysis*

As shown above, the methodology is based on a combination of two allocation approaches (EPC and GF). In this section, we conduct a sensitivity analysis to compare results using two other combinations (Table 1). The first is equal per capita and final consumption expenditures (EPC + FCE) in which the assigned share per capita is proportional to the final consumption expenditure. We therefore used the average final consumption expenditures on education in Canada (other than universities, between 2010 and 2020) divided by total expenditures for the period [32] to calculate the share of the carrying capacity assigned to schools, which is about 1.26%. The second is equal per capita and economic value added (EPC + EVA), where the assigned share is proportional to the economic value added (share of elementary and secondary schools' GDP [33] to the national GDP, which is about 3% [34].

**Table 1.** Overview of allocation approaches used to compare results.

| Sharing Principle | Person Share | Person Share per Building Construction | Student Share | Floor Area Share |
|---|---|---|---|---|
| EPC + FCE | Equal per capita $(1/\text{POP\_t})$ | FCE $\text{FCE}_{Sc}/\text{FCE}_{CAN}$ | Time spent $h_{Sc}/h_{year}$ | Building floor area $m^2{}_{Sc,CAN}/m^2{}_{B,CAN}$ |
| EPC + EVA | | EVA $\text{GDP}_{Sc}/\text{GDP}_{CAN}$ | | |

*2.5. Application to the Use of Steel in Schools*

In this section, we estimate the quantity of steel that a school with a steel structure and a school with wood structure consume over their life cycles (Appendix E). The school has an area of 4000 m². Data on a building's steel components with predominant steel assemblies and predominant wood assemblies are taken from [35]. The Appendix E includes the details on the different building components. In Canada, the average lifespan of a steel structure building is 77.3 years versus 51.6 for a wood structure building [36]. Building components have different lifespans. According to CLF (2018), the average lifespans of roofs, partition walls and exterior cladding, windows and doors is about 30 years. An HVAC system has a lifespan of about 20 years.

**3. Results**

*3.1. Growth in World Demand*

Figure 3 shows that the future demand for materials is projected to increase rapidly between 2020 and 2060 and slows from 2060 to 2100. This is explained by the fact that the global population growth levels off in the second half of the century. Results show that there is higher demand for iron, which is estimated to reach 30 billion tonnes by 2100, followed by aluminum (300 million tonnes), manganese (80 million tonnes), copper (47 million tonnes), diatomite (30 million tonnes), perlite (7 million tonnes) and molybdenum (1 million tonnes). Although the regression analysis results show an increase in demand for all materials, the annual percentage of growth over 2020–2060 varies between materials: 6% on average for iron, 3% for aluminum, manganese and perlite and 2% for copper. The calculated increases in metal production to meet the anticipated demand reflect the medium growth of two factors: population and income. However, this growth could change with other factors, such as the evolution of the energy mix, as shown in [37], and which was not modeled in the present study. For example, green energy will lead to the enhanced use of solar power, wave power and other renewable energy sources, promoting a shift to some degree to higher demand for copper and aluminum for enhanced grid energy distribution and iron-upgraded energy infrastructures [37]. Ref. [38] found that the various low-carbon energy systems scenarios are more metal-intensive than traditional energy systems, thus leading to higher growth. In contrast, initiatives to meet climate targets through building and infrastructures could increase the use of secondary resources (e.g., steel scrap) and

reduce the production of primary materials [39]. The fraction of estimated adapted users each year differs from one material to the next, depending on the substituability index: iron has the smallest fraction (around 0.3% annually), followed by manganese (around 0.6% annually) and copper (more than 1% annually). The accumulated fractions in time until 100% represent the adaptation times. We did not account for the adaptation for aluminum, perlite, diatomite or molybdenum because adaptation will start after the year 2100, owing to their large reserves.

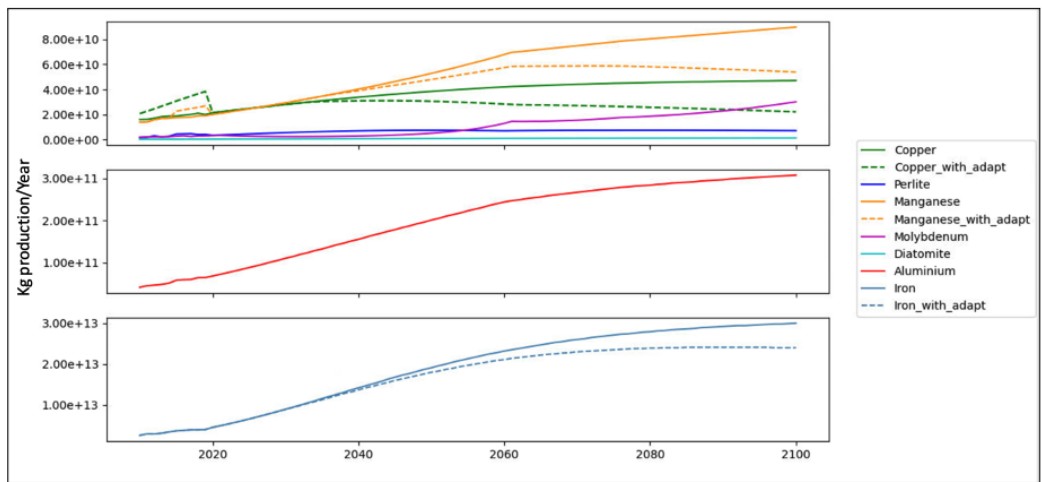

**Figure 3.** Global demand forecast according to SSP2 (output of regression analysis) without accounting for adaptation (solid lines) and accounting for adaptation (dashed lines).

The output of the regression analysis (Appendix A) using the other four SSPs displays different projections (Figure 4). For most materials, high future production is associated with SSP3, which corresponds to low economic growth and high population growth. By 2100, copper reaches 60 million tonnes, iron 61 billion tonnes, aluminium 0.411 billion tonnes and perlite 26 million tonnes. The lowest production is associated with the SSP1: high economic growth but low population growth. By 2100, copper reaches 22 million tonnes, iron 3.6 billion tonnes, aluminum 0.12 billion tonnes and perlite 0.13 million tonnes.

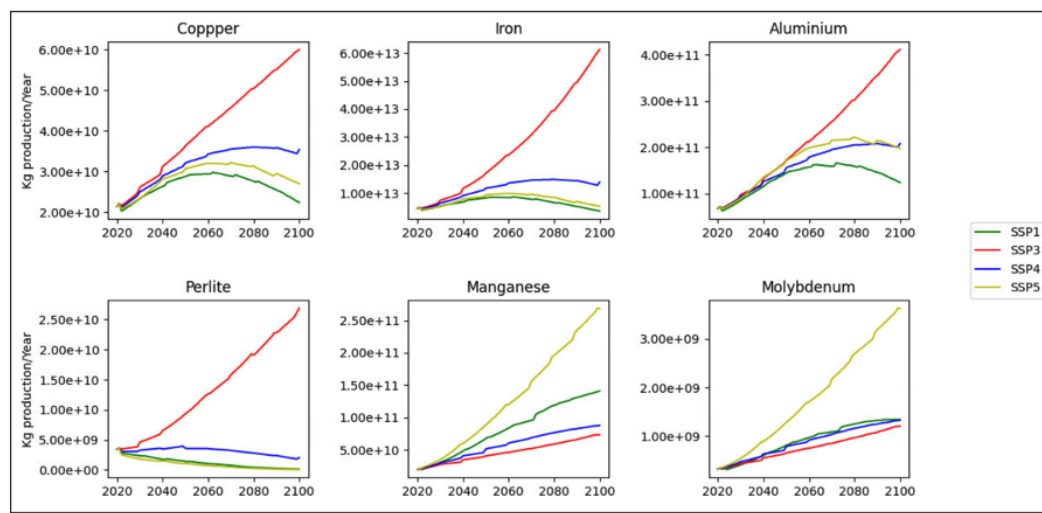

**Figure 4.** Global demand forecast for copper, iron, aluminum, perlite, manganese and molybdenum according to other socioeconomic pathways (SSP1, SSP3, SSP4, SSP5).

### 3.2. World Dissipation Rate vs. World Dissipation Carrying Capacity

Iron, copper, and manganese show an exceedance of $WDR_{CC}$. The gap between $WDR$ and $WDR_{CC}$ increases over time due to demand growth Figure 5. Even though 80% of steel inputs to the economy are from recycling, demand growth contributes to complete dissipation within 89 years, while the fraction of non-adapted users remains high (78%) (Appendixes B and C). Moreover, in some applications, it is very difficult to substitute steel with another material because of its strength and cost-effectiveness [29]. The relatively high adaptation time results in a low iron $WDR_{CC}$ (Figure 5). This explains why the cumulative $WDR$ is 70% higher than the cumulative $WDR_{CC}$. The average $WDR$ of steel is about 3870 million tonnes; however, the average $WDR_{CC}$ is about 264 million tonnes. This means that the $WDR$ of steel is 14 times higher than the estimated $WDR_{CC}$. The diminution of the $CC$ could be achieved trough the diminution of the adaptation time through the development of other alternatives to fulfill the steel functions.

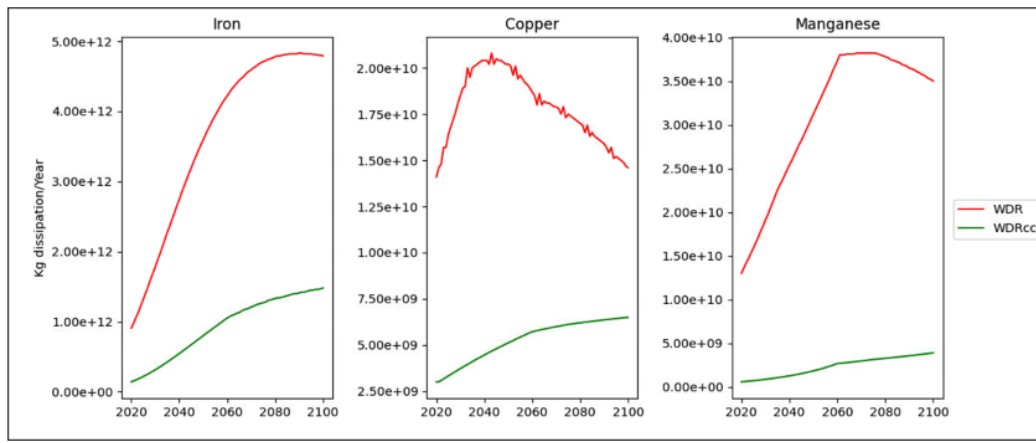

**Figure 5.** Comparison of the world dissipation rates ($WDR$) of iron, copper, and manganese with the calculated carrying capacity ($WDR_{CC}$).

In the case of copper, more than 30% of the annual supply is from recycled sources and nearly all copper products may be recycled repeatedly without loss in product properties. Results show that around 15 million tonnes of copper are dissipated annually (Figure 5). In addition, it is estimated that complete dissipation will occur within the next 90 years; 40% of users are still non-adapted. This is explained by the unique properties of copper in terms of thermal and electrical conductivity, which make it difficult to substitute. The actual dissipation ($WDR$) exceeds the carrying capacity by 56%. Dissipating less than the carrying capacity ($WDR_{CC}$) (less than 6 million tonnes annually) of copper would maintain the reserve until users have totally adapted. In the case of manganese, as an iron alloy, it follows iron in all steel applications. The annual recovery rate of manganese is estimated at 37% [40] and has no satisfactory substitutes in its major applications [29]. The actual $WDR$ exceeds the $WDR_{CC}$ by around 68%, and the entire dissipation of the reserve is predicted for 90 years from 2020, when around 54% of users will still not be adapted.Finally, the high recovery rate of the aluminum (80%) substitute availability (different composites, steel, vinyl or wood can substitute for aluminum), as well as the large reserve base, makes it sufficiently available for future use. With the same calculations for iron for different SSPs Figure 6, the SSP1 pattern shows that, due to a decline in production as of 2076, the carrying capacity exceeds the dissipation of iron. However, the cumulative dissipation until 2100 exceeds the carrying capacity by 22.4% for SSP1, 79.7% for SSP3, 54.8% for SSP4 and 32.6% for SSP5.

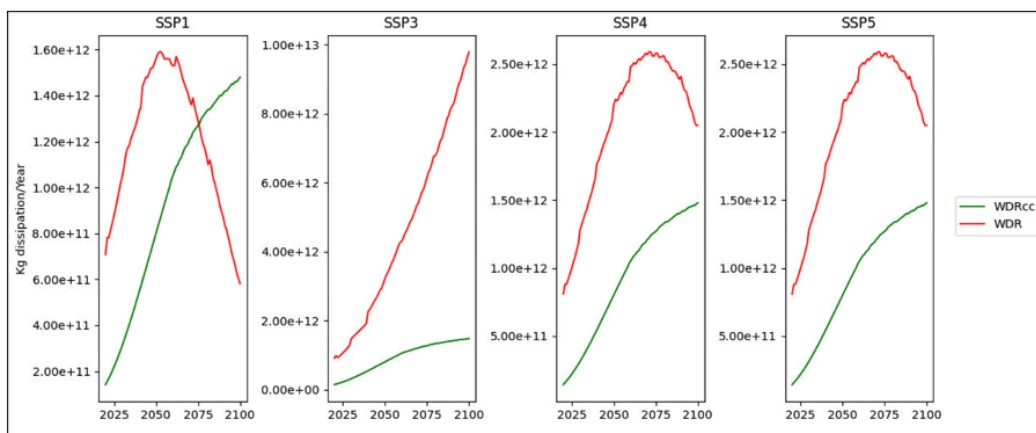

**Figure 6.** Projections of iron world dissipation rate (*WDR*) versus world dissipation rate carrying capacity (*WDR_{CC}*) (2022–2050) according to SSP1, SSP3, SSP4 and SSP5.

### 3.3. Assigning Carrying Capacity to the Elementary School Level

Figure 7 presents the assignment of a fraction of the carrying capacity to building construction. Iron is the most commonly used metal worldwide in construction. Up to 26% of iron is used in construction (as steel). The use of steel in building construction covers the most important components, such as foundations, basements and frames, and is hardly substitutable in terms of performance (substitution index = 1 and $t_a dapt \approx 1000$ years). For copper, while around 28% of the quantity produced is used annually for building construction [26], it can easily be substituted by plastic, stainless steel and aluminum, depending on the application, and that explains the substitutability index of 0.3 and the predicted decline of copper demand in construction as of 2060. Manganese is largely used in the building construction sector as the main material in steel metallurgy. Its substitutability index for the construction sector is equal to 1, which means that there are no available alternatives to manganese for this function. Today, the allocated dissipation to schools is estimated at 2.74 kg per student or 0.45 kg per m$^2$ for steel, 0.025 kg per student or 0.004 kg per m$^2$ for copper, 0.11 kg per student or 0.018 kg per m$^2$ for manganese, and 0.09 kg per student or 0.016 kg per m$^2$ for aluminum. Allocation according to student number is 80% higher than allocation according to square meter of the floor area because the fraction of time a student spends in school is much higher than the fraction of m$^2$ that schools share to total Canadian buildings. The classroom is an average of 50 m$^2$ and holds an average of 20 students—a higher density than other building types [41].

### 3.4. Sensitivity Analysis

In this section, we evaluate how applicable different allocation approaches to share dissipation are to a building type and how sensitive the share is to the choice of allocation principle. Figure 8 compares three allocation approaches: EPC + FCE, EPC + EVA and EPC + GF. The grandfathering approach allocates the higher dissipation carrying capacity in the case of iron. This is explained by the high mass of iron currently used in building construction compared to the use of copper and manganese. However, in the case of final consumption expenditures and equivalent value, we used constant factors that we applied to all materials without distinction. Results show that EVA is slightly higher than the final consumption expenditure. In the case of a school's contribution to GDP, the indicator provides a measure of the proportion of national wealth invested in educational institutions by linking public and private expenditures with the gross domestic product (GDP). While we consider the sum of household final expenditures and capital final expenditures (construction and maintenance) in the case of FCE, this is not necessarily the general government final consumption expenditure and therefore does not reflect the total investment in schools.

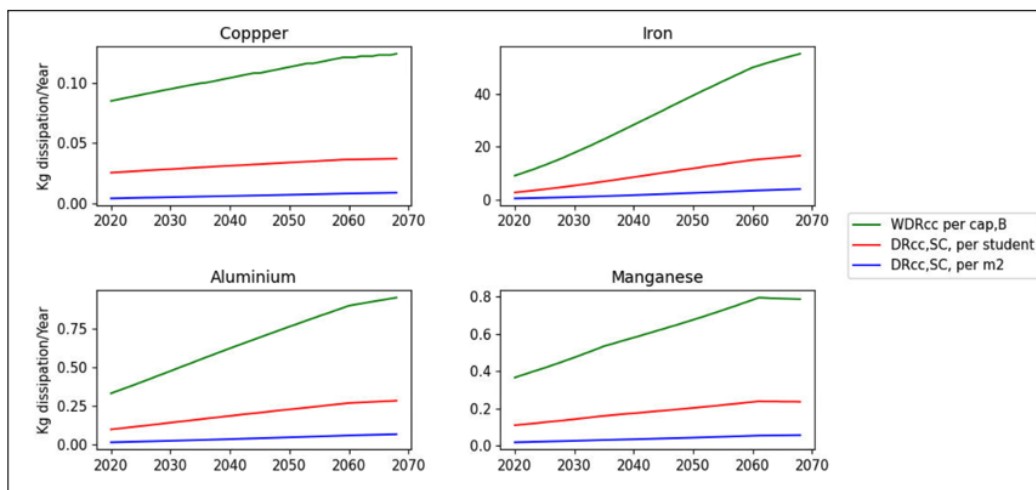

**Figure 7.** Allocation of the world dissipation rate per capita to the building construction sector ($WDR_{CC,per.cap,B}$) and share per student ($DR_{CC,per.student}$) and per school square meter of floor area ($DR_{CC,Sc,per.m^2}$)

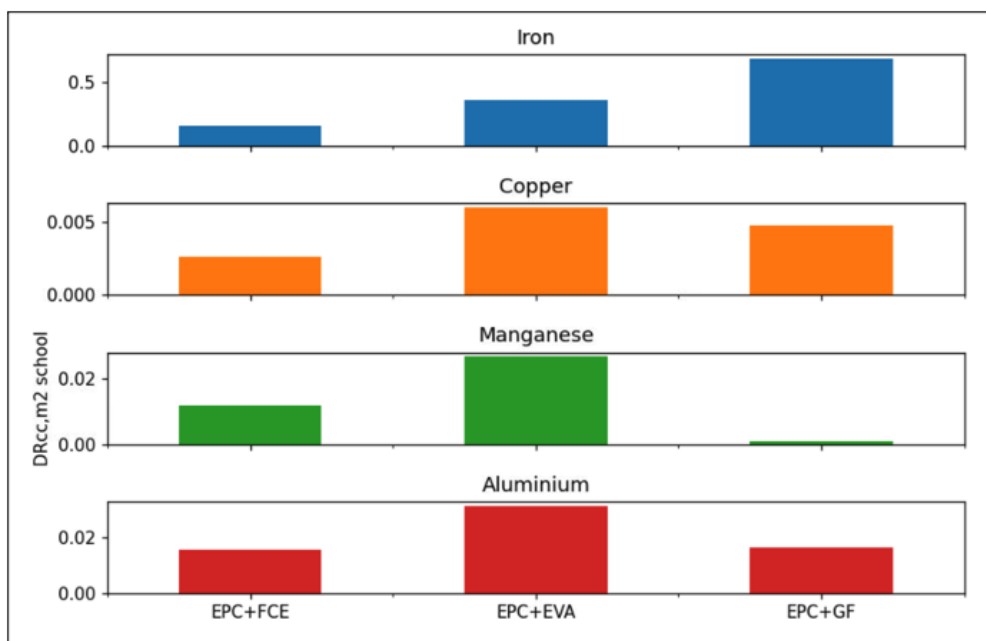

**Figure 8.** Results for the assigned dissipation rate per m$^2$ of a school ($DR_{CC,Sc,per.m^2}$) using different allocation approaches: equal per capita combined with grandfathering (GF), equivalent value added (EVA) and final consumption expenditures (FCE). Data are for the year 2022.

### 3.5. Application to Steel Consumption in a School Life Cycle

Results show that a steel-based structure building consumes 647.85 tonnes (105.56 kg/m$^2$) of steel in its lifespan Figure 9. A wood-based structure building consumes 2.69 tonnes of steel (4.6 kg/m$^2$). The dissipation of steel in steel and wood structures types is 14.3 kg/m$^2$·year (2.86 kg dissipation/m$^2$·year) and 4.56 kg/m$^2$·year (0.09 kg dissipation/m$^2$·year). Our assigned carrying capacity through grandfathering is 2 kg dissipated/m$^2$·year, on average. That means that a steel structure school exceeds the CC by 43%. However, the wood structure is 95% below the CC. Taking into account the lifespan differences of the structures, the comparison for a 77.3-year lifespan (which is 50% longer than the lifespan of a wood structure) increases the dissipation of the wood structure to 0.13 kg·m$^2$·year, which is still 93% below the CC. In addition, the longer the life span of the building, the less the dissipation rate.

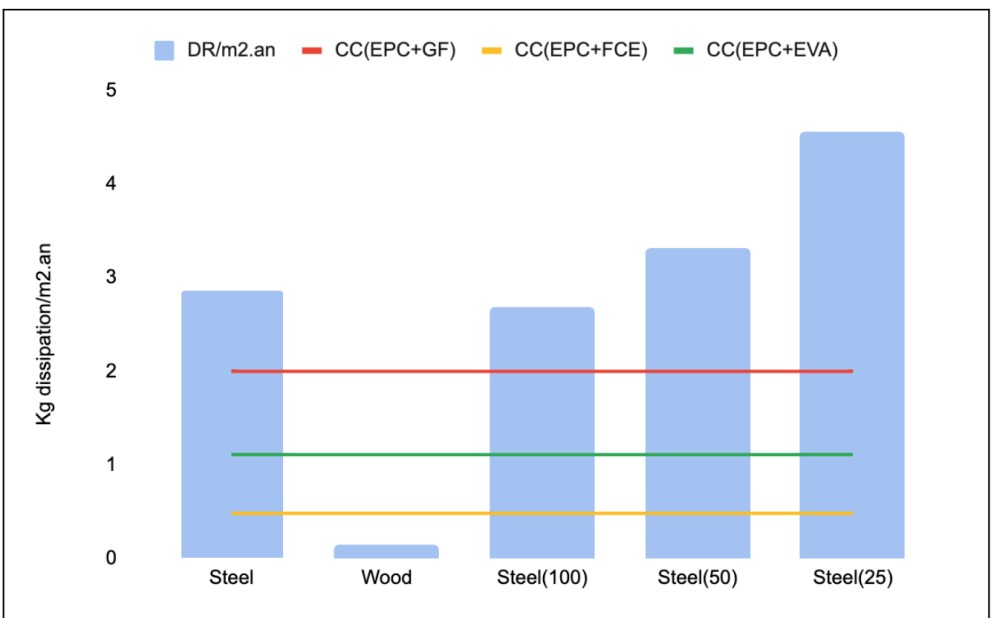

**Figure 9.** Steel dissipation in wood and steel structure types versus the estimated carrying capacity with different allocation principles and under different lifespan periods (100–50–25).

## 4. Discussion

The current dissipation of iron, copper, and manganese are exceeding the estimated dissipation CC. In the case of iron, the future projection shows that only according to SSP1, the CC increases beyond the dissipation from the year 2075. The projection followed the IPAT equation. Some studies estimate the future demand for specific materials through logistic growth curves using a growth factor and a year of production peak, for example in [42] for future lithium production. The studies follow the pattern of petroleum production examined by [43] under the peak oil concept. However, the production peak followed by a decline in production due to resource depletion remains controversial among supporters and detractors alike [44].

Results highlight the importance of reducing dissipation for resources that show a high exceedance of the CC, such as iron. This could be achieved through developing other alternatives or through increasing the reuse and recycling. The case study of the school shows that the wood structure uses the iron sustainably, which means that the design is also influencing the absolute environmental sustainability of the building.

The study uses the model of MACSI because it is a functionality-based model that links the impact from a resource use to its functions and to the availability of alternatives rather than the extraction. However, the MACSI has some limitations. The main limitation is associated with the definition of the adaptation times that is not a robust method. In addition, the substitutability of a material could change with other perspectives, such as cost and technical performance.

As a summary, the advantages of the CC method are as follows:

- Providing a new perspective to the absolute assessment of the impact of resource use and advancing research in the field.
- Estimating a CC based on the estimated evolution of materials demand for a socioeconomic pathway scenario.
- Exploring the concept of instrumental value and dissipation of resources rather than extraction.

The main disadvantages of the CC method are as follows:

- The limit of application to a large number of materials because of the need for a specific data, such as a reserve base.

- The future demand of the material (production) does not include other important drivers, such as technological development, resource efficiency, circular economy or climate change policies. We included only population growth and GDP as drivers for materials production.

In addition, the model does not take into account local habits and design conditions, the rapid market changes and the impact of possible war conflicts in a given region of the world. Those are important factors that could be included in the impact modeling of resource depletion. The future availability of a resource is a function of supply restriction due to geological, technical, environmental, social, political and economic factors that define a material criticality but are not considered in this study [45].

As most LCIA models, the current approach cannot be calibrated or checked, but aims at assessing a potential impact. Here, what we consider "a potential impact" is dissipating the resources at a higher rate than the one that would allow all the users to adapt before the depletion occurs, based on the best estimate we could have of the dissipation rate, the stocks and the adaptation capacity of the users. As all LCIA models, it aims at enlightening decision making and not at measuring an impact.

## 5. Conclusions

The purpose of the current study is to define a CC for abiotic resource dissipation based on a functionality-based model and the projection of future production. The CC was used, for the first time, to explore the AESA of building sector for this impact category. When applied to the case of steel use in a steel-based structure of a Canadian school, we found that the dissipation exceeds the dissipation CC, even when using different allocation approaches, and increases the school lifespan.

In practice, the method could be applied in the building sector at different scales depending on the allocation level. An allocation per construction sector allows at macro level to manage resources according to the CC allocated to the sector. The federal government and building association are more involved at this level. An allocation per building life cycle allows to manage the resource use at the building scale or per type of building. Architects and certification bodies are involved at this scale. Additionally, the study may be applied to different building types (hospitals, universities, single-family homes, etc.), or even other sectors, such as industry. The method could be applied to other materials when data are available. Another case study could be carried out, exploiting the findings of this study.

Building upon this study, some recommendations are given to strengthen the concept of the CC of resource dissipation and to adopt it for conducting an AESA at the building sector:

- Deepen research on resource substitutability and future potential development of other alternatives per resource per application/function. This influences mainly the adaptation time that was used in this study to estimate the CC of dissipation.
- Conduct studies that estimate the anthropogenic stock of materials (e.g., in Canadian buildings). As the study considers the CC of dissipation, we could also include resources that remain in use and that have the potential to be available in the future.
- Consider to study also an allocation to building sector per area of need as the need for housing, hospitalization, or education could allow more CC budget than the recreational building.
- Adapt the finding of the study to the LCA results. To do so, an aggregated CC with the same unit as LCA results should be developed.

**Author Contributions:** Conceptualization, W.B., C.B. and A.B.; methodology, W.B.; software,W.B.; investigation, W.B.; writing—original draft preparation, W.B.; writing—review and editing, W.B., C.B. and A.B.; supervision, C.B. and A.B. All authors have read and agreed to the published version of the manuscript.

**Funding:** This work was funded through the NSERC CREATE Heritage program at Carleton University (NSERC grant number: 465459-2015).

**Institutional Review Board Statement:** Not applicable.

**Informed Consent Statement:** Not applicable.

**Data Availability Statement:** Not applicable.

**Conflicts of Interest:** The authors declare no conflict of interest.

## Appendix A. Projection of Future Production

The IPAT (Equation (A1)) expresses $I$, the environmental impact (production) as a function of $P$, population, $A$, affluence (GDP per capita) and $T$, technology (impact per GDP) [26].

$$I = P.A.T \tag{A1}$$

Ref. [46] transformed the IPAT model into a stochastic form of the STIRPAT model, which allows for non-proportional effects and statistical tools to assess the significance of the different drivers (a is a model coefficient, and b, c, and d represent the exponentials of the independent variables) (Equation (A2)).

$$I = aP^b A^c T^d \tag{A2}$$

To simplify driver estimation, ref. [46] converted the model into a logarithmic form (regression model). In the absence of any data on the technological evolution, a proxy assumption that technology is constant was used (d logT = constant). The components (d logT) and (log a) are replaced by coefficient C (Equation (A3)).

$$LogI = C + b.logP + c.logA \tag{A3}$$

In addition, the environmental impact is assumed to reflect demand (production): I is $WPR_{projected}$, and affluence is the GDP per capita (Equation (A4)).

$$LogWPR_{projected} = C + b.logP + c.log(GDP/cap) \tag{A4}$$

## Appendix B. Regression Analysis Results

**Table A1.** SSP2.

| Materials | C | b | c | $p$-Value |
|---|---|---|---|---|
| Iron | −41.65 | 5.46 | 0.05 | 0.02 |
| Copper | −9.98 | 1.99 | 0.13 | 0.005 |
| Aluminum | −23.17 | 3.22 | 0.51 | 0.00071 |
| Manganese | −8.9 | 1.45 | 1.7 | 0.4 |
| Perlite | −41.89 | 5.82 | −1.49 | 0.1 |
| Diatomite | 113.8 | −14.05 | 8.42 | 0.005 |
| Molybdenum | 113.78 | 14.05 | 8.42 | 0.3 |

**Table A2.** SSP1.

| Materials | C | b | c | $p$-Value |
|---|---|---|---|---|
| Iron | −39.74 | 5.22 | 0.17 | 0.02 |
| Copper | −9.986 | 1.99 | 0.13 | 0.005 |
| Aluminum | −23.17 | 3.22 | 0.51 | 0.00067 |
| Manganese | −8.9 | 1.45 | 1.14 | 0.1 |
| Perlite | −41.33 | 5.7 | −1.49 | 0.1 |
| Molybdenum | −11.86 | 1.69 | 0.88 | 0.4 |

**Table A3.** SSP3.

| Materials | C | b | c | *p*-Value |
|---|---|---|---|---|
| Iron | −39.62 | 5.20 | 0.19 | 0.02 |
| Copper | −9.986 | 1.98 | 0.144 | 0.005 |
| Aluminum | −23.11 | 3.21 | 0.528 | 0.00071 |
| Manganese | −9.23 | 1.5 | 1.12 | 0.4 |
| Perlite | −41.13 | 5.72 | −1.44 | 0.1 |
| Molybdenum | −13.11 | 1.83 | 0.83 | 0.3 |

**Table A4.** SSP4.

| Materials | C | b | c | *p*-Value |
|---|---|---|---|---|
| Iron | −39.62 | 5.20 | 0.19 | 0.02 |
| Copper | −9.94 | 1.98 | 0.144 | 0.005 |
| Aluminum | −23.11 | 3.21 | 0.528 | 0.00071 |
| Manganese | −8.9 | 1.5 | 1.12 | 0.4 |
| Perlite | −41.13 | 5.72 | −1.44 | 0.1 |
| Molybdenum | −9 | 1.31 | 1.1 | 0.4 |

**Table A5.** SSP5.

| Materials | C | b | c | *p*-Value |
|---|---|---|---|---|
| Iron | −39.62 | 5.2 | 0.19 | 0.02 |
| Copper | −9.986 | 1.99 | 0.144 | 0.001 |
| Aluminum | −23.11 | 3.21 | 0.528 | 0.00071 |
| Manganese | −9.2 | 1.5 | 1.12 | 0.4 |
| Perlite | −41.13 | 5.72 | −1.44 | 0.1 |
| Molybdenum | −9.5 | 1.38 | 1.04 | 0.09 |

## Appendix C. Materials Substitutes/User Adaptation

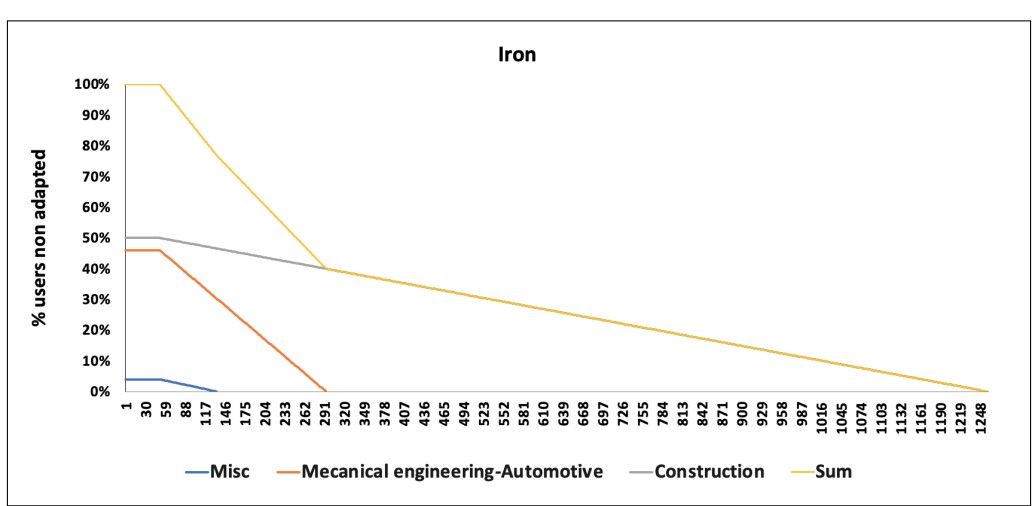

**Figure A1.** Dynamic of users adaptation for iron.

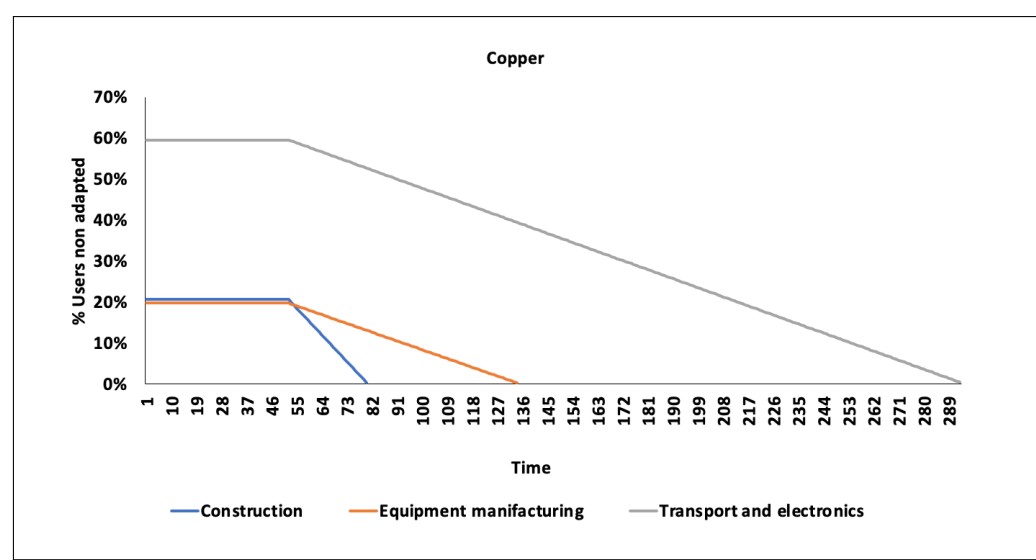

**Figure A2.** Dynamic of users adaptation for copper.

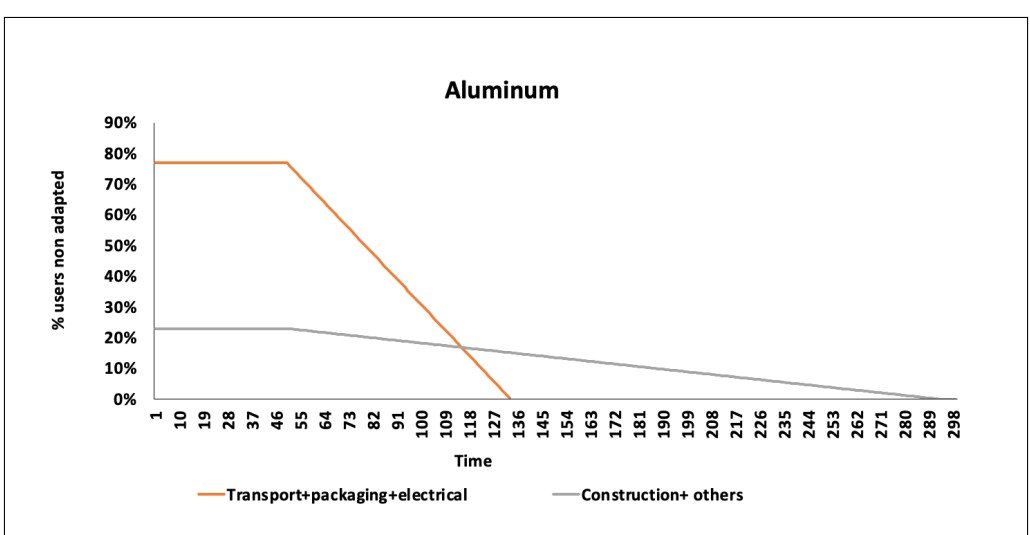

**Figure A3.** Dynamic of users adaptation for aluminum.

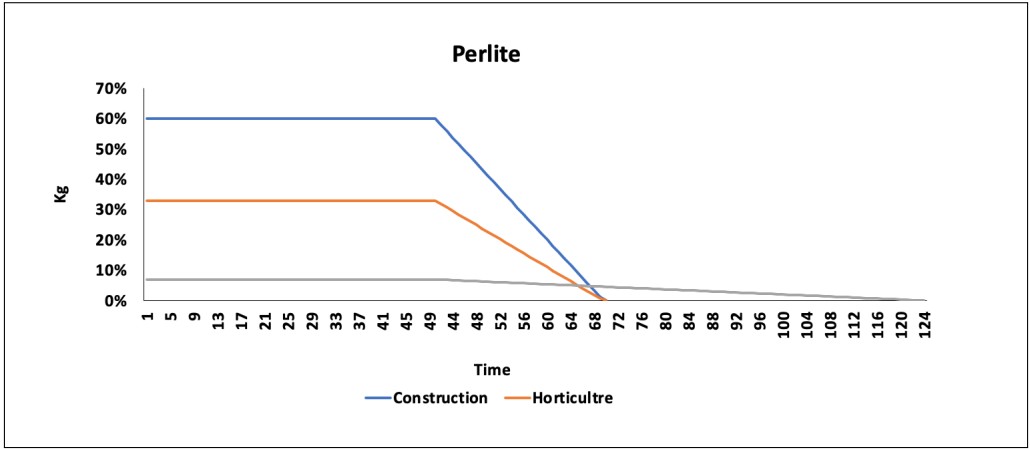

**Figure A4.** Dynamic of users adaptation for perlite.

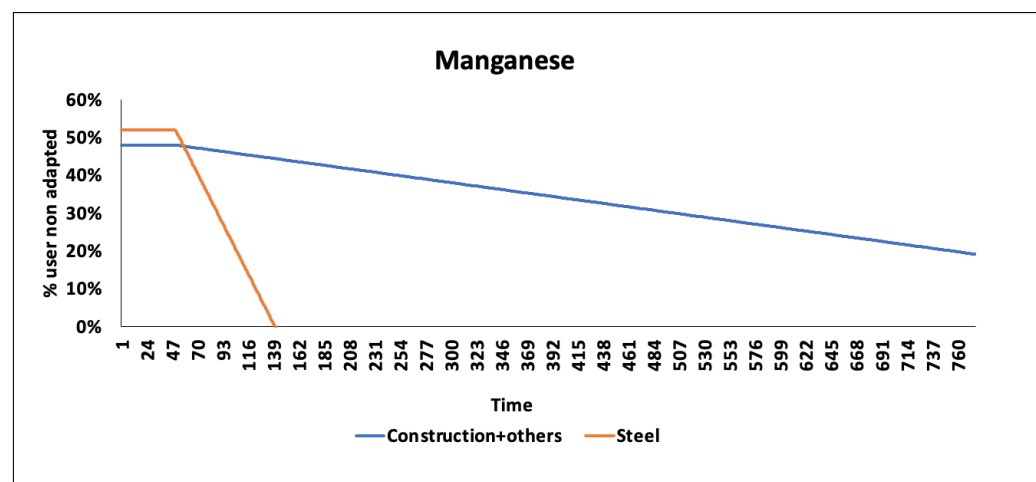

**Figure A5.** Dynamic of users adaptation for manganese.

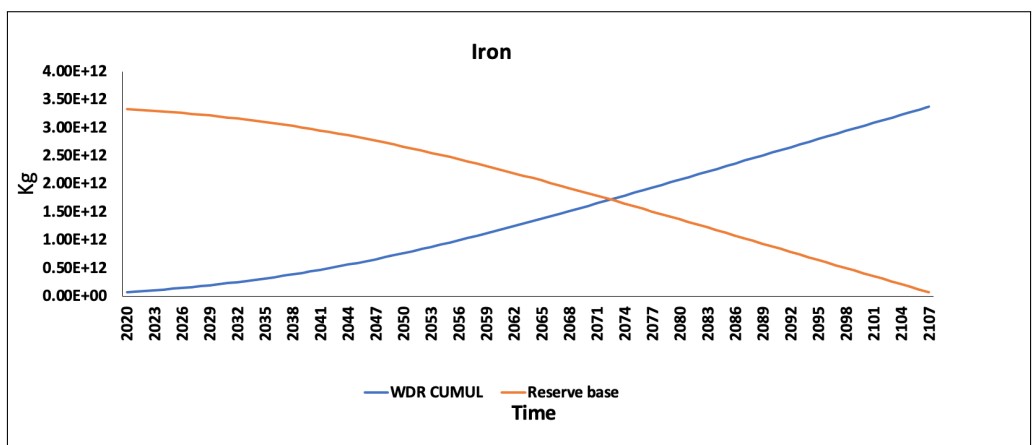

**Figure A6.** The evolution of the world dissipation rate cumul (*WDR*) and the reserve base from 2020 to 2100 for iron.

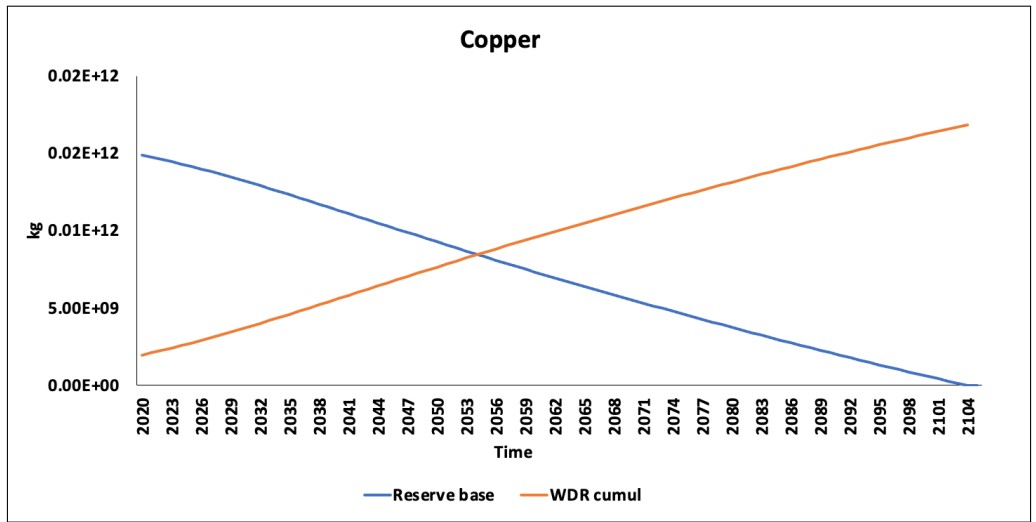

**Figure A7.** The evolution of the world dissipation rate cumul (*WDR*) and the reserve base from 2020 to 2100 for copper.

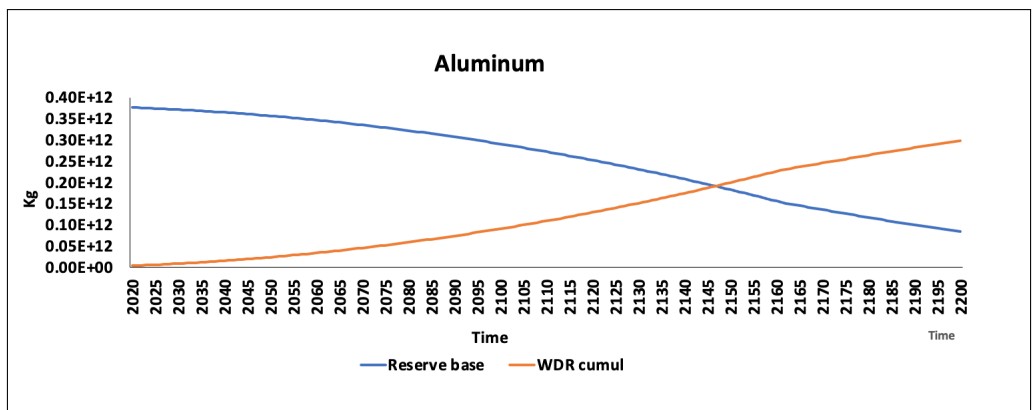

**Figure A8.** The evolution of the world dissipation rate cumul (*WDR*) and the reserve base from 2020 to 2100 for aluminum.

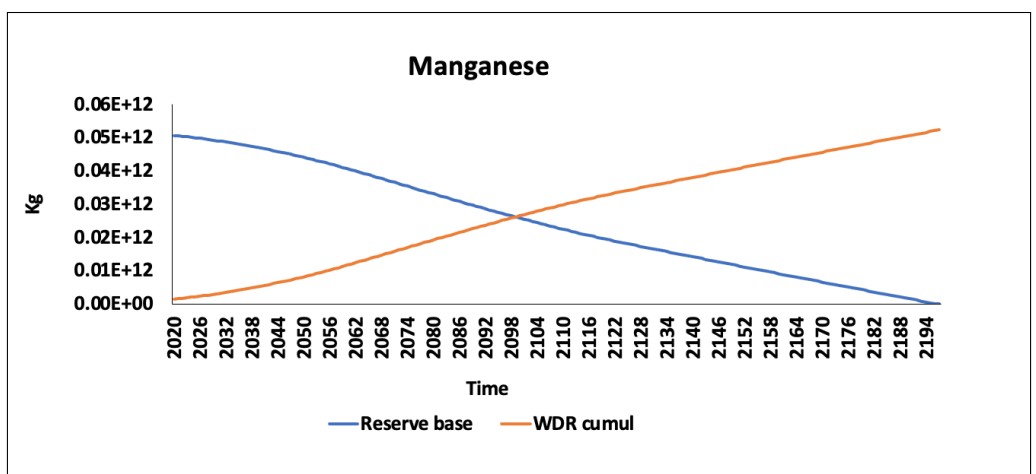

**Figure A9.** The evolution of the world dissipation rate cumul (*WDR*) and the reserve base from 2020 to 2100 for manganese.

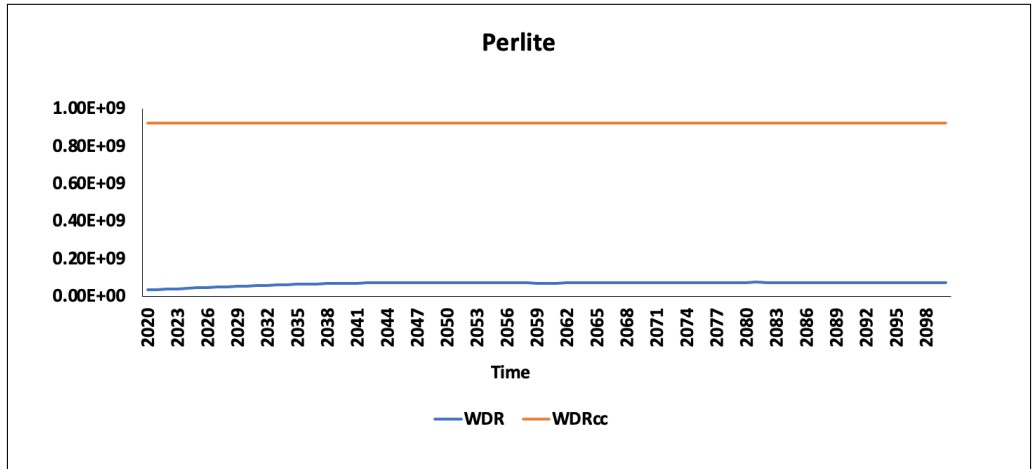

**Figure A10.** The evolution of the world dissipation rate cumul (*WDR*) and the reserve base from 2020 to 2100 for perlite.

## Appendix D. Additional Literature about the Allocation of the Carrying Capacity to Building Sector

**Table A6.** Summary of Studies using Allocation Principles in the Building Sector.

| Study | Building Type | Location | Environmental Concern | Allocation to Country | Allocation to Building/Building Type | Absolute Sustainability Assessment |
|---|---|---|---|---|---|---|
| [19] | Detached houses | New Zealand | CC | EPC | GF climate impact. | NA |
| [20] | Residential buildings. | Denmark | TA-TE-WD-LUS-LUB-CC-OD-FE-EP-POF-FET | EPC | EVA and GF | LCA of a standard house and an upcycled single-family house. |
| [21] | Colleges and universities | USA | CC | EPC | GF AND Time spent | NA |
| [18] | Different designs of single-family dwelling | Denmark | CC-ME-LU-WRD | Multiple allocation approaches | Multiple allocation approaches | LCA was conducted for six dwellings. |
| [47] | Detached, medium-density housing and apartments | New Zealand | CC | EPC | GF | Case study per building typology and future projections |
| [48] | Residential building | Czechia | CC | EPC | GF | LCA study for a four-storey residential building with different improvement measures. |

CC: Climate change; EPC: equal per capita; GB: global budget; GF: grandfathering; TA: terrestrial acidification; TE: terrestrial eutrophication; WD: water depletion; LUS: land use soil erosion; LUB: land use biodiversity; OD: ozone depletion; FE: freshwater eutrophication; EP: marine eutrophication; POF: photochemical oxidant formation; FET: freshwater ecotoxicity; LU: land use; I/O: input output.

## Appendix E. Additional Data Used for the Case Study

**Table A7.** Steel Quantities (Estimated from [35]).

| Building Component | Steel Structure | Quantities (Tonnes) | Wood Structure | Quantities (Tonnes) |
|---|---|---|---|---|
| Ceiling and roof | 39mm × 0.76 mm galvanized metal deck 550 | 4395.13 | | |
| Mezzanine floors | 39 mm × 0.76 mm galvanized corrugated | 9.22 | 0.53 mm galvanized steel resilient @ 600 mm o/c | 2.4 |
| Ceiling and roof | 39mm × 0.76mm galvanized metal deck 550 | 4395.13 | | |
| Partition walls | 39 mm × 152 mm × 0.91 mm steel studs @ 400 mm o/c | 1.98 | | |

**Table A7.** *Cont.*

| Building Component | Steel Structure | Quantities (Tonnes) | Wood Structure | Quantities (Tonnes) |
|---|---|---|---|---|
| Structural system | Beams: 350 MPa W-sections Columns: 350 MPa hollow structural steel (H.S.S.) sections Lateral Bracing: Steel rod x-bracing hot-rolled steel connection plates, fasteners, and misc. steel | 10.32 | Lateral Bracing: Steel rod x-bracing hot-rolled steel connection plates, fasteners, and misc. steel | 0.36 |
| Exterior walls | 339 mm × 152 mm 1.21 mm steel studs @ 400 mm o/c | 2.5 | 1.21 mm heavy-duty galvanized steel furring channels@ 400 mm (self-weight: 0.82 kg/m) | 1.56 |
| Foundation | 150 mm × 150 mm × 3.4 mm steel mesh reinforcement @ 0.886 kg/m$^2$ | 3.55 | 1150 mm × 150 mm × 3.4 mm steel mesh reinforcement @ 0.886 kg/m$^2$ | 3.55 |
| Doors | insulated steel exterior door ( 813 mm × 2134 mm) Steel interior doors with no glazing (813 × 2134 mm) | 6.66 | | |
| HVAC system | Steel | 0.37 | Steel | 0.37 |

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
