# Peer review of "Absolute Environmental Sustainability of Materials Dissipation: Application for Construction Sector"

_resources, doi:10.3390/resources11080076_

Round 1
Reviewer 1 Report
The work is descriptive, it can be described as a theoretical analysis.
In the opinion of the Reviewer, the paper concerns the estimation (prediction) of economic parameters.
Therefore, the possibility of publishing the above paper as related to construction raises doubts, it certainly does not qualify for recognition as a technological article, it is certainly not an article related to structural elements, and it is certainly not an article describing the theoretical analysis of structural systems. It can be concluded that the paper submitted for review relates to the issues of construction economics.
In the opinion of the Reviewer for the construction industry, the paper submitted for review is of no significant importance. Perhaps it is of such importance for economic and environmental issues as well as environmental protection.
The article requires quite significant changes if it is to be published:
1) lack of information about the advantages and disadvantages of the CC method and the MACSI used,
2) how was the applied research method calibrated,
3) how is it known that the obtained results are reliable, how was the correctness of the obtained results checked,
4) is the model used universal or can it be used to analyze the degree of wear of other building materials?
5) whether the model can be used in different parts of the world, does it take into account local habits and design conditions
6) whether the model takes into account the possibility of rapid market changes and the impact of possible war conflicts in a given region of the world on the obtained results of theoretical analyzes.
The conclusion that wear and tear increases with increasing durability of the object is obvious and nothing new.
To sum up, the paper requires additions and details, after introducing these changes, another attempt to publish it can be made.
Reviewer 2 Report
The authors propose to the scientific community and practitioners, a very timely issue: stock assessment of resources by providing a tool to monitor the impact of their depletion. The article is very well written with a clear and rigorous approach. The bibliography cited is up-to-date and of high quality.
I would recommend some improvements to the authors before publication.
(1) The study is valuable, however, the authors should better clarify to the readers of the journal the innovative contribution of their research. What knowledge gaps does it address? Highlighting this aspect would enhance the article! In addition, the inclusion of one or more research questions could help make the research objective more substantive.
(2) In the conclusion, I suggest that the authors highlight the theoretical contribution that their resource use impact assessment model makes to knowledge about impact assessment systems such as LCA. Reading the article or inferred that the proposed method could solve some limitations of LCA analysis, but this aspect is not clear in the manuscript and would like to be stressed.
(3) Also, it would be appropriate to emphasize how the use of this model has implications for practitioners. Finally, do the authors believe that their approach could be applied to other sectors besides construction? I am thinking, for example, of natural resource-intensive industries such as ceramics and glass are, by the way, both of which are closely related to the building industry.
Author Response
.
